# Establishment and Characterization of Immortalized Miniature Pig Pancreatic Cell Lines Expressing Oncogenic K-Ras^G12D^

**DOI:** 10.3390/ijms21228820

**Published:** 2020-11-21

**Authors:** Hae-Jun Yang, Bong-Seok Song, Bo-Woong Sim, Yena Jung, Unbin Chae, Dong Gil Lee, Jae-Jin Cha, Seo-Jong Baek, Kyung Seob Lim, Won Seok Choi, Hwal-Yong Lee, Hee-Chang Son, Sung-Hyun Park, Kang-Jin Jeong, Philyong Kang, Seung Ho Baek, Bon-Sang Koo, Han-Na Kim, Yeung Bae Jin, Young-Ho Park, Young-Kug Choo, Sun-Uk Kim

**Affiliations:** 1Futuristic Animal Resource & Research Center, Korea Research Institute of Bioscience and Biotechnology, Cheongju-si 28116, Korea; baboi87@kribb.re.kr (H.-J.Y.); sbs6401@kribb.re.kr (B.-S.S.); embryont@kribb.re.kr (B.-W.S.); icaros1019@kribb.re.kr (Y.J.); unebin@kribb.re.kr (U.C.); leedg@kribb.re.kr (D.G.L.); cjj728@kribb.re.kr (J.-J.C.); bsj@kribb.re.kr (S.-J.B.); dvmlim96@kribb.re.kr (K.S.L.); lhy3650@kribb.re.kr (H.-Y.L.); son1989@kribb.re.kr (H.-C.S.); gt1300@kribb.re.kr (P.K.); 2Department of Biological Science, College of Natural Sciences, Wonkwang University, 460, Iksan-daero, Iksan-si 54538, Korea; 3National Primate Research Center, Korea Research Institute of Bioscience and Biotechnology, Cheongju-si 28116, Korea; choiws@kribb.re.kr (W.S.C.); ck2816@kribb.re.kr (S.-H.P.); nemo9426@kribb.re.kr (K.-J.J.); bsh82@kribb.re.kr (S.H.B.); porco9@kribb.re.kr (B.-S.K.); hnkim@kribb.re.kr (H.-N.K.); ybjin@gnu.ac.kr (Y.B.J.); 4Department of Laboratory Animal Medicine, College of Veterinary Medicine, Gyeongsang National University, 501 Jinjudaero, Jinju 52828, Korea; 5Department of Functional Genomics, KRIBB School of Bioscience, Korea University of Science and Technology (UST), Daejeon 34113, Korea

**Keywords:** pancreatic ductal adenocarcinoma, acinar-to-ductal metaplasia, K-ras^G12D^, miniature pig pancreatic cells

## Abstract

In recent decades, many studies on the treatment and prevention of pancreatic cancer have been conducted. However, pancreatic cancer remains incurable, with a high mortality rate. Although mouse models have been widely used for preclinical pancreatic cancer research, these models have many differences from humans. Therefore, large animals may be more useful for the investigation of pancreatic cancer. Pigs have recently emerged as a new model of pancreatic cancer due to their similarities to humans, but no pig pancreatic cancer cell lines have been established for use in drug screening or analysis of tumor biology. Here, we established and characterized an immortalized miniature pig pancreatic cell line derived from primary pancreatic cells and pancreatic cancer-like cells expressing K-ras^G12D^ regulated by the human PTF1A promoter. Using this immortalized cell line, we analyzed the gene expression and phenotypes associated with cancer cell characteristics. Notably, we found that acinar-to-ductal transition was caused by K-ras^G12D^ in the cell line constructed from acinar cells. This may constitute a good research model for the analysis of acinar-to-ductal metaplasia in human pancreatic cancer.

## 1. Introduction

Pancreatic cancer is a severe type of human cancer, with a 5-year survival rate of 5–10%, because it lacks specific symptoms and is difficult to diagnose during the early stages [1,2]. Pancreatic cancer occurs primarily in exocrine components, and fewer than 5% of all tumors occur in endocrine cells [3]. Pancreatic ductal adenocarcinoma (PDAC) constitutes more than 90% of exocrine malignancies. PDAC precursor lesions include proliferating epithelial lesions, pancreatic intraepithelial neoplasia (PanIN), and prominent lesions [4]. The main cause of PanIN formation is the activation of mutated K-ras in >90% of patients with PDAC. This activation is caused primarily by G12D amino acid substitution [5]. Progression from PanIN to PDAC is reportedly related to the accumulation of mutations in tumor suppressor genes such as p53, p16, SMAD family member 4 (SMAD4) (, and BRCA1 DNA Repair Associated 1/2 (BRCA1/2) [6].

The exocrine pancreas, consisting of acinar cells and ducts, accounts for >90% of the organ mass. However, only 10% of the ductal cells were previously presumed to be cells of PDAC origin due to the duct shape of most PDACs [7]. However, recent studies have suggested that acinar cells are more sensitive to K-ras mutation and tend to progress from PanIN to PDAC via acinar-to-ductal metaplasia (ADM) [8,9]. It is generally thought that PanIN originates from acinar cells, while intraductal papillary mucinous neoplasms are derived from ductal cells [10,11]. Nevertheless, these lesions can co-exist. This complexity requires further elucidation to understand the roles of these precursor lesions in the initiation and progression of PDAC.

Mice have been widely used as experimental animals in pancreatic cancer research due to the simplicity and low costs of breeding management and genetic modification [12,13]. However, mice have limitations in relation to human diseases, whereby they do not reflect the anatomical and physiological characteristics of humans [14]. Moreover, only 5–8% of anticancer drugs that have entered clinical trials from preclinical studies are approved for clinical use [15]. Therefore, large animals have received considerable attention as potential alternative models of pancreatic cancer [16,17].

Miniature pigs are useful for biomedical research because they share many similarities with humans in terms of body size, organ size and structure, physiology, and pathophysiology [18,19]. Cancer biology research involving miniature pigs is a new field, but there is evidence that cancers in pigs accurately mimic cancers in humans. Sequencing of the porcine genome and the simultaneous advancement of gene editing techniques have led to research in porcine models of cancers, including colorectal, breast, liver, and pancreatic cancers [20,21]. In 2015, the oncopig model was described. This transgenic pig possesses the K-ras^G12D^ and p53^R167H^ oncogenic transgenes under the control of Cre-recombinant proteins [22]. The oncopig is now commercially available, and additional transgenic pig models are expected in the near future. These models may be useful for investigation of pancreatic cancer.

In this study, we identified changes in various genes causing a cancer-like phenotype in miniature pig pancreatic cells (PCs) expressing oncogenic K-ras^G12D^. In addition, we observed whether the acinar-to-ductal transition involves K-ras^G12D^, as mentioned above. To investigate this aspect, we generated immortalized PCs in the pancreatic tissue of miniature pigs and then constructed a cell line expressing K-ras^G12D^ controlled by the acinar cell-specific pancreas-specific transcription factor 1 A (PTF1A) promoter. Comparison with immortalized PCs may provide insights regarding the action of K-ras^G12D^ in the pancreas and could yield important information for the development of subsequent pancreatic cancer pig models, as well as provide a marker for early detection of pancreatic cancer. To the best of our knowledge, this is the first study to demonstrate the in vitro development of porcine pancreatic tumor-like cells required for early analysis of pancreatic cancer.

## 2. Results

### 2.1. Generation of a Miniature Pig PC Line Expressing K-ras^G12D^ under the Control of the PTF1A Promoter

To determine whether mutated human K-ras (K-ras^G12D^) expression leads to K-ras-driven tumorigenic phenotypes in miniature pig PCs (Figure 1A), we first isolated primary PCs from miniature pig pancreatic tissue. However, PCs failed to grow within 3 days after plating, adopting an enlarged flattened morphology, and they expressed senescence-associated β-galactosidase expression on day 6. Therefore, we induced cellular transformation by the transduction of pBABE-puro SV40-LT into PCs, creating an immortalized PC line (iPCs). Unlike PCs, iPCs demonstrated indefinite cell growth extension and did not undergo senescence (Appendix A). Next, we induced expression of mutant K-ras under the human PTF1A promoter, which is important for expression in PCs. This was achieved by linking the 3-kb human PTF1A promoter to the hemagglutinin (HA)-tagged human K-ras^G12D^ gene and enhanced green fluorescent protein (EGFP) with an internal ribosome entry site (IRES) (Figure 1B). We established three independent stable transgenic cell lines expressing K-ras^G12D^ (KiPCs) from parental iPCs and then used the exogenous K-ras^G12D^-expressing cell line (#1) with the strongest proliferation for this study (Appendix A). Semi-quantitative reverse transcription polymerase chain reaction (PCR) showed that the exogenous K-ras^G12D^ and EGFP were expressed in KiPCs, whereas they were not expressed in other samples from pancreatic tissues, PCs, or iPCs (Figure 1C). In accordance with this, Western blot and immunocytochemistry data confirmed that protein levels of exogenous K-ras^G12D^ and EGFP were detected only in KiPCs (Figure 1D,E). Therefore, our data indicated that the human PTF1A promoter is sufficient to regulate the expression of human oncogenic K-ras in an SV40-LT-mediated immortalized miniature pig PC line.

### 2.2. Cellular Morphology and Growth Characteristics of Miniature Pig KiPCs

All cells grew in a monolayer tightly attached to the bottom of the cell culture flasks. However, the morphological patterns of these cells consisted of different shapes and sizes under phase-contrast microscopy. PCs exhibit a flattened or elongated multipolar fibroblast-like shape and somewhat different sizes, although most cultures contained predominately large cells. iPCs were polygonal in shape with more regular dimensions and smaller sizes but included fewer spindle-shaped cells (i.e., a typical fibroblast shape). In contrast, KiPCs typically showed a tightly packed cell morphology, comprising small round or polygonal epithelial-like cells with a cobblestone pattern (Figure 2A). We evaluated whether the morphological characteristics of these three cell types were remarkably different with respect to size (forward scatter) or internal complexity (side scatter) using flow cytometry. Immortalized cell lines (iPCs and KiPCs) were identified by a single-cell sorting process, which separated the transfected cells into individual wells, and performed subsequent subculture (Appendix A). Therefore, PCs contained multiple cell populations, whereas iPCs and KiPCs were relatively homogeneous (Figure 2B). Consistent with the flow cytometry data, single cells showed considerably smaller mean lengths in cultures of immortalized cells compared with PCs (Figure 2C). We next assessed whether the morphologic characteristics induced by oncogenic K-ras^G12D^ reflected cell growth. As shown in Figure 2D,E, the growth rates of the cell lines were slow and did not significantly differ during the first 2 days. Subsequently, KiPCs grew more rapidly than PCs and iPCs until day 7 (Figure 2D). There was a noticeable difference in growth rate among the cell types at day 6 (Figure 2E). The results were similar in anchorage-independent growth experiments (Appendix A). Little or no cell growth was observed in PCs and iPCs, whereas KiPCs demonstrated both significantly more cells and significantly larger cells. Taken together, the findings suggested that oncogenic K-ras^G12D^ induces accelerated growth and altered morphology in PCs.

### 2.3. Altered Expression Patterns of Cell Cycle and Apoptosis-Associated Genes in KiPCs

To clarify changes in the acceleration of proliferation in KiPCs, we analyzed the DNA content by flow cytometry and the expression profile of genes related to cell-cycle progression among the cell lines. First, we found distinct differences among the cell types in terms of the cell cycle. PCs showed cell-cycle arrest at the G0/G1 phase, whereas the proportion of cells in the G0/G1 phase significantly decreased from 98.9% to 88.2% in iPCs and to 75.1% in KiPCs. Conversely, the proportion of S-phase cells significantly increased in KiPCs (23.9%), which was a greater increase than that in iPCs (8.15%) (Figure 3A). In accordance with this, the number of Bromodeoxyuridine (BrdU)-positive cells was also significantly increased in KiPCs compared with PCs and iPCs (Figure 3B). To further validate these data, we analyzed the levels of cell cycle-related genes in PCs, iPCs, and KiPCs by quantitative PCR (qPCR) and Western blotting. We performed the screening of cell-cycle regulatory genes such as p14, p15, p16, p18, p19, p21, and p27 (i.e., cyclin-dependent kinase inhibitors). Unexpectedly, only the level of p18 was significantly decreased, whereas that of p21 was increased, in iPCs compared with PCs and KiPCs. However, the levels of all cyclin-dependent kinase inhibitors were most reduced in KiPCs compared with PCs and iPCs (Figure 3C). Moreover, the levels of cyclin-related and proliferation-related genes were also significantly increased in KiPCs compared with iPCs. As shown in Figure 3D, compared with iPCs, mRNA levels of cyclins (B1, D1, and E1) and cyclin-dependent kinases (1), (2), and (4) were upregulated in KiPCs. KiPCs also demonstrated a proliferative state, which was characterized by the expression of proliferating cell nuclear antigen and c-myc (Figure 3D). These results were similar at the protein level and were more robust in KiPCs than in iPCs, confirming that the proliferation of cells was further accelerated by oncogenic K-ras^G12D^.

Based on the results of previous studies [23], we investigated whether the expression of K-ras^G12D^ influences the mitogen-activated protein kinase (MAPK) signaling pathway in KiPCs, using Western blotting to measure the levels of proteins phosphorylated by MAPK kinase. The results revealed that c-Jun N-terminal kinases (JNK) activation was not significantly affected in KiPCs, whereas the levels of activated phosphor extracellular-signal-regulated kinase 1/2 (ERK1/2), pAKT, and pp38 were significantly higher in KiPCs than in PCs or iPCs (Figure 3F). Next, we investigated whether K-ras^G12D^ promotes cell proliferation by regulating iPC apoptosis. Notably, significant differences in the mRNA levels of anti-apoptotic genes (BCL-2 and BCL-XL) and pro-apoptotic genes (BAK and BAX) were observed in KiPCs compared with iPCs (Figure 4A). Consistent with this, Western blot and Terminal deoxynucleotidyl transferase-mediated dUTP-digoxigenin nick end-labeling (TUNEL) analyses showed a reduction in expression and the proportion of apoptotic cells (Figure 4B,C). Therefore, KiPCs were presumed to have acquired cancer-like properties via changes in related genes and phenotypes (e.g., promotion of cell growth and reduction of apoptosis).

### 2.4. Robust Induction of Pancreatic Ductal and Cancer Cell Markers in KiPCs

Previous studies have demonstrated the importance of K-ras^G12D^ expression in acinar cells for the progression of ADM lesions to PanIN and PDAC [24]. Here, we investigated whether our acinar cells undergo transdifferentiation into duct-like cells (i.e., acinar-to-ductal transition) and whether our cells initiate the transgenic cascade, thereby fully mimicking human disease. We confirmed the purity of the starting acinar cell preparation by analyzing the expression of representative PCs lineage markers (acinar, ductal, and endocrine) using qPCR. In particular, the expression of acinar markers (SYP, CTRP1, PTF1A, PAX4, and PAX6) was relatively increased in iPCs (Figure 5A). However, a marked shift from acinar to ductal gene expression was observed in KiPCs lines (#1, #6, and #7). We found that the expression of ductal markers (CA2, CK19, and SOX9) was increased, whereas that of acinar markers was significantly reduced, in KiPCs compared with iPCs (Figure 5A). Consistent with the qPCR data, the CK19 protein in KiPCs was observed by immunofluorescence analysis (Figure 5E). These results indicated that acinar-to-ductal transition occurred in KiPCs after the induction of oncogenic K-ras^G12D^.

To investigate whether KiPCs acquired early cancer cell features based on their cancer-like properties (e.g., proliferative phenotype and gene expression), we assessed the expression levels and patterns of pancreatic cancer markers. Compared with PCs and iPCs, the gene expression levels of some pancreatic cancer markers (epithelial cell adhesion molecule (EpCAM), CD44, CD133, and c-MET) were significantly increased in KiPCs (Figure 5C). In addition, immunocytochemical analysis of the association between endogenous EpCAM expression and reporter EGFP labeling further supported that KiPCs are cancer-like cells (Figure 5F). After treatment with MAPK kinase inhibitors, the pattern of lineage markers was analyzed to also determine whether acinar-to-ductal transition and pancreatic cancer related genes was related to MAPK kinase in KiPCs. As a result, ductal (CA2, CK19, and SOX9)/pancreatic cancer (EpCAM and CD44) markers decreased and acinar markers (SYP and PTF1A) increased by inhibitor treatment in KiPCs (Figure 5B,D). Taken together, these data suggested that KiPCs exhibit ductal phenotype and cancer-like characteristics in response to oncogenic K-ras^G12D^ signaling as a mediator of ADM.

## 3. Discussion

Recently, many studies have been performed to understand and treat pancreatic cancer. Among these, mechanistic studies have been conducted using mouse models [25,26,27]. Despite the importance of mouse models, these models do not fully reflect human disease. Moreover, they have limitations such as differences in size, macroscopic pancreatic organization, and immunity [17,28]. Therefore, pancreatic cancer research requires a new animal model that is more similar to humans. Relative to mice, pigs are more similar to humans [29] in terms of anatomy, physiology, and immunology. Diseases affecting the pancreas are likely better represented in pig models [19]; thus, pigs may provide a more clinically relevant model of PDAC [30]. However, despite the importance of a porcine model for pancreatic cancer, there are few pig PC lines available for pancreatic studies.

In the current study, we produced SV40-LT-mediated iPCs and then used these cells to generate KiPCs expressing acinar-specific K-ras^G12D^, using a single-cell culture method [31,32]. The data presented herein demonstrated the newly acquired characteristics of our cells mediated by changes in genes related to proliferation and lineage markers. These cells are presumably good research material for pancreatic cancer research. KiPCs exhibited notable characteristics. The proportion of acinar cells was expected to increase based on human PTF1A promoter usage, but the expression levels of ductal markers were increased, while those of acinar markers were relatively reduced, by K-ras^G12D^. In addition, the expression of the PTF1A gene was reduced in KiPCs compared with the parental iPCs (Figure 5A). Previous studies have shown that the loss of PTF1A, a master regulator of acinar differentiation, is sufficient to induce ADM and is dramatically sensitive to K-ras^G12D^-induced transformation [33]. ADM is a complex process that is considered the first step in the development of pancreatic cancer induced by oncogenic K-ras^G12D^, substantially accelerating the development of PDAC via abnormal inflammatory and growth factor signaling through cell surface receptors [34,35]. The MAPK pathway is activated by mutant K-ras^G12D^ in pancreatic cancer [36,37]. Upon activation of the Ras pathway, wild-type Ras binds to guanosine triphosphate and regulates various downstream signaling molecules. When the bound guanosine triphosphate is released, the RAS pathway is inactivated [38]. However, the G12D mutation in Ras leads to constitutive guanosine triphosphate binding and continuous activation of the RAS pathway, thus activating the phosphoinositide 3-kinase (PI3K)/AKT, ERK1/2, c-Jun N-terminal kinases (JNK), and p38 signals [38,39]. Here, we confirmed that ERK1/2, AKT, and p38 were activated in KiPCs (Figure 3F). Therefore, the importance of the MAPK signaling pathway is supported by its promotion of PanIN, which renders ADM susceptible to transformation [40]. A recent study suggested that the NF-κB inhibitor interacting Ras-like protein (κB-Ras) affects Ral activity to regulate the downstream Ras signaling pathway [41]. Notably, low levels of κB-Ras are expressed in patients with PDAC, and κB-Ras deficiency promotes ADM. Nevertheless, the mechanisms of ADM progression and transformation remain unclear because physiological factors have not been fully elucidated.

The currently available porcine pancreatic cancer cell lines (i.e., PGKP, PGKPS, and PGKPSC) were generated from primary pancreatic ductal cells obtained from domestic pigs via the expression of oncogenic K-ras^G12D^ and p53^R167H^, with or without knockdown of p16^Ink4A^ and SMAD4 [30]. Remmers et al. reported that these three transformed PCs acquired tumorigenicity characteristics such as increased proliferation, soft agar colony formation, and invasion in vitro, as well as induced tumorigenesis in nude mice. Another PC line was produced by inducing Ad-Cre into primary pancreatic ductal cells obtained from the oncopig model (LSL-K-ras^G12D^/p53^R167H^ cassette) developed in 2015, and cancer cell characteristics were reported in that model [42]. However, these pancreatic cancer cell lines are derived from pancreatic ductal cells and thus are most useful for evaluating intraductal papillary mucinous neoplasms. To our knowledge, our iPCs and KiPCs are the first pig pancreatic acinar cancer cells, which may constitute ideal models for analysis of ADM.

Mouse models of pancreatic cancer are successful in terms of pancreatic cancer formation but have several limitations, which prompted the development of a pig model of pancreatic cancer. A recent study demonstrated the production of transgenic pigs with Cre-inducible K-ras^G12D^ and p53^R167H^ regulated by the CAG promoter [30,42]. Ad-Cre delivery into pancreatic ducts led to tumor formation using adenoviruses. However, this led to both pancreatic and neuroendocrine tumors, presumably due to the non-specific viral transduction and constitutive activity of the CAG promoter. In addition, clinical features were not evident, and tumors were identified by resecting the pancreatic duct at 1 year after adenoviral injection. Pancreatic cancer formed, but its value as a pancreatic cancer model was reduced because the human clinical phenotype did not appear, and tumors were formed at sites distant from the pancreas. Previous reports have described the production of a pancreatic cancer pig model using a cassette designed to express K-ras^G12D^, c-Myc, and SV40-LT genes under the control of the mouse pancreatic duodenal homeobox 1 (Pdx1) promoter [43]. Most of the newborn mice died early, and tumors had formed throughout the pancreas. The Pdx1 promoter was presumed to have been expressed throughout the pancreas, because the Pdx1 promoter plays a crucial role in the differentiation of endoderm into pancreatic endoderm during embryonic development [44]. Despite tumor formation throughout the pancreas, there was no clear evidence of ADM.

We created a miniature pig model of pancreatic cancer using the hPTF1A promoter–HA-K-ras^G12D^–IRES–EGFP vector to discover specific markers of PanIN as a potential therapeutic targeted against mutant K-ras^G12D^ (Appendix A). As expected, we found that the oncogenic K-ras^G12D^ gene was specifically expressed in the pancreas under control of the human PTF1A promoter in transgenic pigs (Appendix A). We used various image analyses to follow the progression of pancreatic precursor lesions in K-ras^G12D^ founders (F0) transgenic pigs. Preliminary evidence showed that the level of CA 19-9, often used as an indicator of pancreatic cancer, did not differ between wild-type and K-ras^G12D^ F0 transgenic pigs at 12 months of age. Moreover, mass lesions were not detected by ultrasound imaging (Appendix A). However, positron emission tomography/computed tomography showed slightly increased absorption of 18-F fluorodeoxyglucose in the pancreas, and pancreatic abnormalities were confirmed by gross organ analysis (Appendix A). Although the timing of pancreatic cancer formation was slower than in the mutant K-ras^G12D^/p53^R273H^ pig model, and there were differences in tumor progression [30], we consider K-ras^G12D^ transgenic pigs to be a good research model to determine the mechanism and therapeutic approach for targeting K-ras^G12D^ in PDAC. Additionally, when the transgenic founder was sexually mature, we crossed the transgenic founder with wild-type pigs to obtain offspring (F1) generation piglets. Of these piglets, four were transgenic F1 and one was a non-transgenic offspring, showing the possibility of transgene transfer to the germline (Appendix A). We are currently obtaining transgenic F1 piglets by breeding and continuing to follow pancreatic cancer progression in transgenic founders using image analyses. The discovery of characteristic changes in cancer-related genes in KiPCs through further analysis is expected to be useful in establishing better in vitro and in vivo models.

## 4. Materials and Methods

### 4.1. Preparation of Miniature Pig Primary Pancreatic Cells

All procedures and use of male of KRIBB small pig (KSP) miniature pigs were approved by the Korea Research Institute of Bioscience and Biotechnology (KRIBB) Institutional Animal Care and Use Committee (Approval No. KRIBB-AEC-16075). All surgical procedures were performed under anesthesia, and all efforts were made to minimize animal suffering. For euthanasia, pigs were injected with ketamine (5 mg/kg intravenously, Yuhan, Seoul, Republic of Korea) followed by KCl injection (75–150 mg/kg, Dai Han Pharm. Co, Chungcheongbuk-do, Republic of Korea), and all efforts were made to minimize animal suffering by institutional veterinary experts, in accordance with National Institutes of Health (NIH) guidelines. A 2-week-old KSP miniature pig [45] was provided by the Futuristic Animal Resource & Research Center in KRIBB, Korea. Primary pancreatic cells (PCs) were prepared from the miniature pig pancreas as preciously described [46] and maintained in RPMI 1640 medium (#11875119, GIBCO, Grand Island, NY, USA) supplemented with 10% fetal bovine serum (FBS, #16000-044, GIBCO), 100 U/mL penicillin and 100 µg/mL streptomycin (#15140122, GIBCO). Cells were incubated in a 37 °C incubator supplemented with 5% CO_2_, and non-PC cells were removed by 3 cycles of incubation for 20 min according to their slower adhesion onto flask surface. Add MAPK inhibitor, SP600125 (#420119, Calbiochem, La Jolla, CA, USA), PD98059 (#513000, Calbiochem), U0126 (#662005, Calbiochem), SB203580 (#559389, Calbiochem), and LY294002 (#440202, Calbiochem) to RPMI 16,400 medium and culture 48 h.

### 4.2. Generation of Miniature Pig Pancreatic Cell Lines

The hPTF1A promoter-HA-K-ras^G12D^-IRES-EGFP-hygro expression vectors were modified from pCAGGS-hygro, pBMN-IRES-GFP (#1736, Addgene, Cambridge, MA, USA), and pBabe-K-ras^G12D^ (#58902, Addgene) respectively, by replacing the CMV early enhancer/chicken β actin (CAG) promoter between Sal I and Nhe I restriction sites with human PTF1A 3kb promoter, which was PCR amplified from human genomic DNA.

To immortalized primary miniature pig PCs (iPCs), PC cells were transfected with pBabe-puro SV40 LT (#13970, Addgene) retrovirus. After 48 h, cells were treated with 5 μg/mL puromycin for selection. iPCs were transfected with the hPTF1A promoter-HA-K-ras^G12D^-IRES-EGFP-hygro vector and selected using 500 µg/mL hygromycin to create KiPC cell lines stably expressing K-ras^G12D^. Subsequently, the cell sorting process is applied to the GFP-labeled single cell by flow cytometer (BD FACSAria™ III, BD Biosciences, San Jose, CA, USA) to obtain 100% clonal purity. Each line was evaluated by Western blotting and immunocytochemistry to confirm efficient overexpression of K-ras^G12D^.

### 4.3. Immunocytochemistry

Cells were grown on Lab-Tek^TH^ II chamber slide^TM^ system (#154526PK, Nunc, Rochester, NY, USA), fixed with 4% formaldehyde (#47608, Sigma-Aldrich, St. Louis, MO, USA) at 4 °C for 24 h, permeabilized with 0.1% triton X-100 in phosphate-buffered saline (PBS, #LB001-02, Welgene, Korea), and blocked with 10% normal goat serum (NGS, #31873, Thermo, Madison, WI, USA) for 1 h at room temperature. Subsequently, antibodies against HA (#11867423001, Roche, Basel, Switzerland), GFP (#Ab290, Abcam, Cambridge, MA, USA), CK19 (#ab84632, abcam), and EpCAM (#PAB283Po01, Cloud-Clone Corp, Wuhan, China)) were incubated with the prepared cells at 4 °C overnight. Finally, the cells were washed several times with PBST (0.02% Tween-20 in PBS) and incubated with Alexa Fluor 594 secondary antibodies (Invitrogen, Carlsbad, CA). Fluorescence was analyzed by fluorescence microscopy (Leica, Wetzlar Germany).

### 4.4. BrdU Incorporation Assay

PC, iPC, and KiPC cell were seed in Lab-Tek^TH^ II chamber slide^TM^ system (#154526PK, Nunc) with 1000s cell per chamber and cultured for 24 h. After 24 h, we treated with BrdU (#ab142567, Abcam) 10 µM/µL for 24 h. The next step was performed the same way as the immunocytochemistry.

### 4.5. TUNEL Assay

To evaluate apoptotic positive cells in PCs, iPCs, and KIPCs, a TUNEL assay was performed using an in situ cell death detection kit (#12156792910, Roche). Cells were seeded in a Lab-Tek^TH^ II chamber slide^TM^ system (#154526PK, Nunc) with 1000 cells per chamber and cultured for 24 h. After 24 h, cells were washed three times in phosphate-buffered saline (DPBS) with 1% (v/v) polyvinyl alcohol(PVA) and fixed in 4% paraformaldehyde overnight at 4 °C. Fixed cell were permeabilized in DPBS containing 0.5% Triton X-100 at RT for 60 min. Non-specific binding sites were blocked by incubation with DPBS containing 10 mg/mL bovine serum albumin (BSA) for 1 h. Subsequently, cells were washed three times with DPBS-PVA and stained with fluorescein-conjugated dUTP and terminal deoxynucleotidyl transferase for 1 h at 38.5 °C. Subsequently, the cells were washed three times with DPBS-PVA and mounted on clean glass slides with 4′,6-Diamidino-2-Phenylindole, Dihydrochloride (DAPI). DAPI-labeled or TUNEL-positive nuclei were observed under a fluorescence microscope (Olympus, Tokyo, Japan). Total and apoptotic cell numbers per field were judged by counting the nuclei with DAPI (blue) and TUNEL (red) signals. Approximately eight-filed per treatment group were used in the TUNEL assays in each independent experiment.

### 4.6. Semi-Quantitative RT-PCR and qPCR

Total RNAs were extracted from cells using the RNeasy plus mini kit (#74136, Qiagen, Hilden, Germany) according to the manufacturer’s instructions. Total RNA (1 μg) was used for cDNA synthesis using ReverTra Ace-α-^®^ (#TOFSK-101, Toyobo, Osaka, Japan). The expression levels of each gene were determined by semi RT-PCR methods. The PCR was carried out in a 20 μL reaction volume using the ExPrime Taq Mater Mix (#G-6000, Genetbio, Korea). Each cycle consisted of a denaturation step at 95 °C for 30 s, an annealing step at 62 °C for 30 s, and an extension step at 72 °C for 30 s. The final extension step was followed by a 5 min extension reaction at 72 °C. Relative expression levels of the genes were measured by real-time RT–PCR using TB Green^®^ Premix Ex Taq™ (#RR420B, Takara Bio, Shiga, Japan) and analyzed with an Mx3000P QPCR systems (Stratagene, La Jolla, CA, USA). The reaction parameters for the qPCR were 95 °C for 5 min, followed by 26–35 cycles of 95 °C for 30 s and 62 °C for 30 s. For the comparative analyses, mRNA expression levels were normalized to Glyceraldehyde 3-phosphate dehydrogenase (GAPDH) and then expressed as the fold-change. The sample delta Ct (SΔCt) value was calculated from the differences between the Ct values of GAPDH and the target genes. The relative gene expression levels between the samples and the controls were determined using the formula 2−(SΔCt−CΔCt). PCR primers for amplification of the pig cDNAs were designed in silico using Primer3 (version 0.4.0, https://bioinfo.ut.ee/primer3-0.4.0) software (Appendix A).

### 4.7. Western Blot Analysis

For the Western blot analysis, 30–60 μg of protein lysates were separated in 8–12% sodium dodecyl sulfate-polyacrylamide gels and transferred onto nitrocellulose membranes (#HAWP04700, Millipore, Billerica, MA, USA). The membranes were incubated overnight with primary antibodies against p38 (#9212, Cell Signaling, Ipswich, MA, USA), pp38 (#4511, Cell Signaling), AKT (#9272, Cell signaling), pAKT (#4060, Cell signaling), ERK1/2 (#9102, Cell Signaling), pERK1/2 (#4370, Cell signaling), JNK (#9252, Cell Signaling), pJNK (#MA5-15228, Thermo scientific), pCDC2 (#sc-136014, Santa Cruz Biotechnology, Santa Cruz, CA, USA), proliferating cell nuclear antigen (PCNA) (#M0879, Dako), c-myc (#9402, Cell signaling), p16 (#ab51243, abcam), p21 (#ab109199, abcam), pp53^ser15^(#AF1043, R&D Systems), CDK4 (#sc-23896, Santa Cruz Biotechnology), Cyclin D1 (#2922 or #2978, Cell signaling), Cyclin E1 (#sc-247, Santa Cruz Biotechnology), Cyclin-dependent kinase 4 (#sc-23896, Santa Cruz Biotechnology), Bcl2 (#sc-7382, Santa Cruz Biotechnology), Bcl-xL (#sc-8392, sc-8392), Bak (#ab199677, abcam), K-ras (#12063-1-AP, Proteintech, Chicago, USA), HA (#11867423001, Roche), GFP (#Ab290, Abcam), and GAPDH (#LF-PA0212, Abfrontier, GAPDH) at 4 °C. Then, the membranes were washed five times with 10 mM Tris–HCl (pH 7.5) containing 150 mM NaCl and 0.2% Tween-20 (TBST) and incubated with horseradish peroxidase-conjugated goat anti-rabbit Immunoglobulin G (IgG) or anti-mouse IgG (1:5000, both from Sigma-Aldrich) for 1 h at room temperature. After washing the blots with TBST, antibody binding was detected using a chemiluminescence detection system (Amersham, Little Chalfont, UK) according to the manufacturer’s instructions.

### 4.8. Cell-Proliferation Assay

Cells were seeded at a density of 1000 cells/well into 96-well cell culture plates. Cells were grown for 7 days. Cell Counting Kit-8 (CCK-8, #CK04, Dojindo, Kumamoto, Japa) solution was added to each well. After 1 h, the absorbance at 450 nm of each well was read on a microplate reader (VERSAmax™, San Jose, CA, USA).

### 4.9. Anchorage-Independent Growth Assay

The anchorage-independent proliferation potential of cells was assessed by plating 3000 cells in 1 mL of RPMI medium with 10% serum containing 0.3% agar and plating in triplicates over a first layer of 0.6% agar in RPMI medium. The cells were grown at 37 °C and 5% CO_2_. After 2–3 weeks, the numbers of colonies were counted. Data represent the means ± SD from three independent experiments performed in triplicate wells.

### 4.10. Flow Cytometry Assay

Cell cycle distribution was determined by DNA staining with PI (#550825, BD Pharmingen, San Diego, CA, USA). A total of 1 × 10^6^ cells was collected and fixed in 70% ethanol. Cell pellets were suspended in PI/RNase staining buffer and incubate for 15 min at room temperature before analysis. The percentages of cells in different phases of the cell cycle were measured using BD FACSAria™ III instrument.

### 4.11. Somatic Cell Nuclear Transfer (SCNT)

SCNT was performed as previously described [47]. Metaphage II (MII) oocytes in PB1 medium (DPBS supplemented with 4 mg/mL BSA, 75 µg/mL penicillin G, and 50 µg/mL streptomycin sulfate) containing 7.5 µg/mL cytochalasin B were cut using a sharp pipette, and then the first polar body and cytoplasm containing chromosomes at metaphase II were removed using the squeezing method under an inverted microscope (DMI 3000B, LEICA) equipped with a micromanipulator (#NT-88-V3, Nikon Narishige, Tokyo, Japan). Donor cells were selected with good refractivity and placed into the perivitelline space. A single cell–oocyte couplet was placed between two parallel electrodes (#CUY 5100-100, Nepa gene, Ichikawa, Japan) and activated by one direct current pulse of 0.24 kV/cm for 50 µs using an Electro Cell Fusion generator in fusion medium consisting of 280 mM mannitol containing 0.1 mM CaCl_2_·2H_2_O, 0.2 mM MgSO_4_·7H_2_O and 0.01% polyvinyl alcohol (PVA, #10981, Sigma-Aldrich), and incubated at 38.5 °C in 5% CO_2_ in air. After 2 h, oocyte–cell couplets that were completely fused as observed under an inverted microscope were selected and activated by one direct current pulse of 1.2 kV/cm for 50 µs in activation medium consisting of 280 mM mannitol containing 0.1 mM CaCl_2_·2H_2_O, 0.2 mM MgSO_4_·7H_2_O, 0.01% PVA, 0.5 mM 4-(2-hydroxyethyl)-1-piperazineethanesulfonic acid (HEPES), and then cultured in post-activation medium, which consisted of in vitro culture (IVC) medium supplemented with 5 mg/mL cytochalasin B and 2 mM 6-dimethylaminopurine, for 4 h at 38.5 °C in 5% CO_2_ in air. After activation, the activated embryos were transferred to IVC medium at 38.5 °C in 5% CO_2_ in air.

### 4.12. ELISA Assay

To evaluate porcine carbohydrate antigen 19-9 (CK19) in serum, we used a serum separator tube and allowed samples to clot for two hours at room temperature before centrifugation for 20 min at approximately 1000× *g*. Then, we assayed freshly prepared serum immediately or stored samples in aliquots at −20 or −80 °C for later use, avoiding repeated freeze/thaw cycles. ELISA Kits were used for analysis of CK19 (#E07C0671, Blue Gene, Shanghai, China) in accordance with the manufacturer’s instructions.

### 4.13. Positron Emission Tomography and Computed Tomography (PET/CT) Image

All PET images were acquired using a PET/computed tomography (CT) scanner (Simense Bigograph-mCT, Malvern, PA, USA). The miniature pigs were initially anesthetized with intramuscular injections of 0.5 mg/kg ketamine (Yuhan, Koreas), maintained with 2% isoflurane (Hana Pharmacy, Korea) in oxygen at a flow rate 2 L/min using an anesthesia machine (Royal Medical, Korea), and immobilized in a supine position in a custom-made bed holder during the PET/CT scan. The level of inhaled CO_2_, O_2_ saturation, pulse, respiration rate, and body temperature were monitored continuously; and body temperature was maintained by a warm blanket. PET imaging with [^18^F] 2-deoxy-2-[fluorine-18] fluoro-D-glucose (18F-FDG) was performed to early detection of cancer. ^18^F-FDG was obtained from a commercial company (DuChemBio Co, Korea).

### 4.14. Ultrasound Image

To detect pancreatic cancer in miniature pigs, performed at 5 months after birth, we used a ESAOTE MYLAB 50 XVISION Ultrasound (Easatos. Co, Genova, Italy) held above the animal in supine position. Anesthesia was induced and maintained in the same way as PET/CT. We placed the miniature pig in an induction chamber until the animal showed no gross movement except for steady respiration. We confirmed adequate anesthesia using the foot-pad reflex test. The miniature pig was placed in a supine position on a dry paper towel covering a heated preparation mat with its nose inside an anesthesia-lined nose cone. With the animal in supine position, we applied a generous layer of ultrasound gel over the entire abdominal area, aiming to minimize trapped air bubbles between the skin and gel and within the gel itself. Then, we placed the probe on the ultrasound gel above the abdomen and orthogonal to the plane of the imaging platform and gently pressed down to visualize the internal organs.

### 4.15. Statistical Analyses

All experiments were repeated at least three times. Data are expressed as the means ± standard deviation (SD). Data were compared using one-way analysis of variance with Holm–Sidak multiple multiple comparisons, using SigmaPlot v13 (San Jose, CA, USA). *p*-values less than 0.05 were considered to indicate statistical significance.

## Figures and Tables

**Figure 1 ijms-21-08820-f001:**
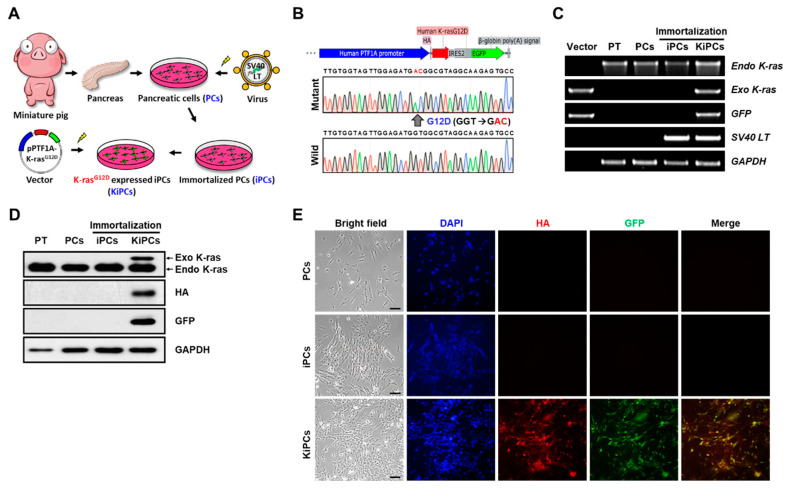
Verification of K-ras^G12D^ expression under the control of the human PTF1A promoter in miniature pig primary PCs. (**A**) Schematic depicting the establishment of K-ras^G12D^ expression in miniature pig PCs. (**B**) Genotyping of the designed K-ras sequence by Sanger sequencing. (**C**) Semi-quantitative reverse-transcription PCR of K-ras, EGFP, and SV40-LT in PT, PCs, iPCs, and KiPCs. (**D**) Protein levels of K-ras, HA tag, and EGFP were measured by Western blot analysis. (**E**) Double immunofluorescence was performed by staining with anti-HA (red) and anti-EGFP (green) antibodies (red fluorescence represents HA-tagged K-ras^G12D^ and green fluorescence Internal Ribosome Entry Site (IRES)-EGFP). Scale bar = 100 µm. Magnification 100×. PT, pancreatic tissue; PCs, pancreatic cells; iPCs, immortalized pancreatic cells; KiPCs, K-ras^G12D^-expressing iPCs; HA, hemagglutinin; EGFP, enhanced green fluorescent protein; PTF1A, Pancreatic-specific transcription factor 1 alpha; SV40-LT, Simian Virus 40 large T antigen.

**Figure 2 ijms-21-08820-f002:**
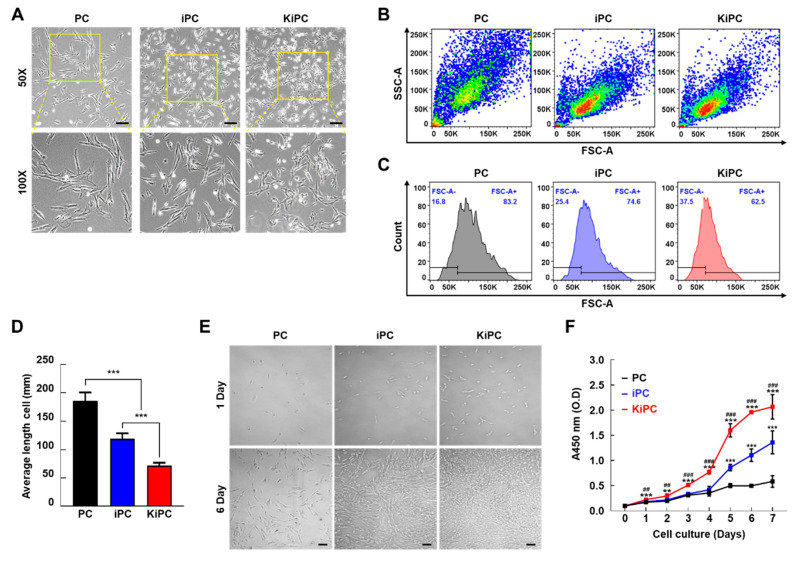
Differences in cellular morphology, size, and proliferation among PCs, iPCs, and KiPCs. (**A**) Representative bright-field images of PCs, iPCs, and KiPCs. Scale bar = 100 µm. Magnification 100× and 200×. (**B**) Flow cytometry analysis of cell populations by forward scatter area (FSC-A) and side scatter area (SSC-A) in PCs, iPCs, and KiPCs. (**C**) Quantitative analysis of FSC-A (cell size) in PCs, iPCs, and KiPCs. Scale is an arbitrary representation of cell size across the entire population. Blue numbers in each figure represent the cell populations in the gated area. (**D**) Lengths of 20 cells in different areas. (**E**) Representative bright-field images of PCs, iPCs, and KiPCs after 1 or 6 days of culture. Scale bar = 100 µm. Magnification 100×. (**F**) Proliferation was measured by CCK-8 assay. Each sample was measured four times at the indicated time points. # denotes a statistically significant difference compared with iPCs; * denotes a statistically significant difference compared with PCs. All experiments were performed using at least three replicates, and the results are representative of three independent experiments. Data are presented as means ± standard deviation. * *p* < 0.05, ** *p* < 0.01, *** *p* < 0.001; one-way analysis of variance with Holm–Sidak multiple comparisons. ## *p* < 0.01, ### *p* < 0.001; one-way analysis of variance with Holm–Sidak multiple comparisons.

**Figure 3 ijms-21-08820-f003:**
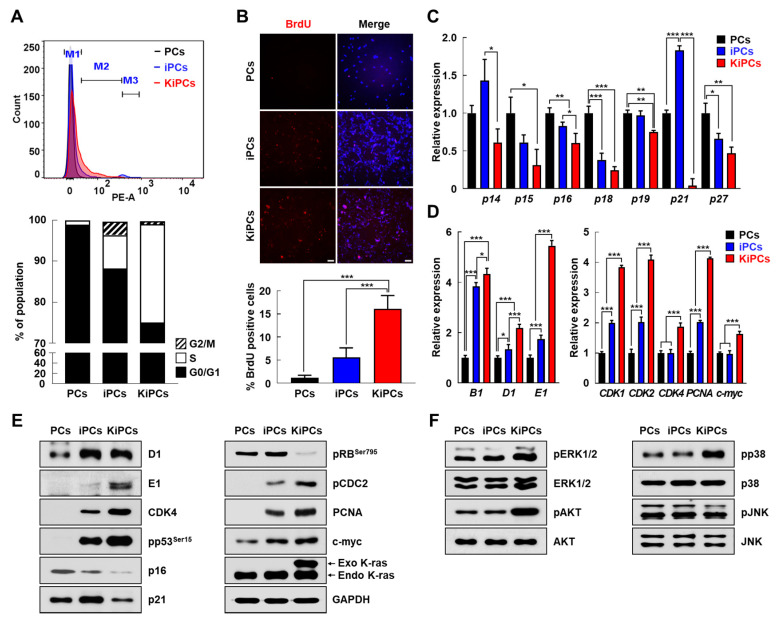
KiPCs exhibited altered levels of cell cycle-regulated genes and mitogen-activated protein kinase (MAPK) kinase pathway members. (**A**) The cell cycle was analyzed by fluorescence-associated cell sorting after DNA staining with propidium iodide. M1: G0/G1, M2: S, and M3: G2/M phase. Percentages of cells in each cell cycle phase are shown in the bar graph. (**B**) Representative images of Bromodeoxyuridine (BrdU) staining (top) and percentages of BrdU-positive cells in the indicated groups (bottom). Magnification 100×. (**C**) qPCR analysis of cyclin-dependent kinase inhibitors. (**D**) qPCR analysis of cyclins and cyclin-dependent kinase-associated genes. (**E**) Cell cycle-related proteins and proliferation markers were measured by Western blot analysis. (**F**) Activation of three MAPKs was measured by Western blot analysis using phosphorylated protein-specific antibodies. Magnification 100×. All experiments were performed using at least three replicates, and the results are representative of three independent experiments. Data are presented as means ± standard deviation. * *p* < 0.05, ** *p* < 0.01, *** *p* < 0.001, one-way analysis of variance with Holm–Sidak multiple comparisons.

**Figure 4 ijms-21-08820-f004:**
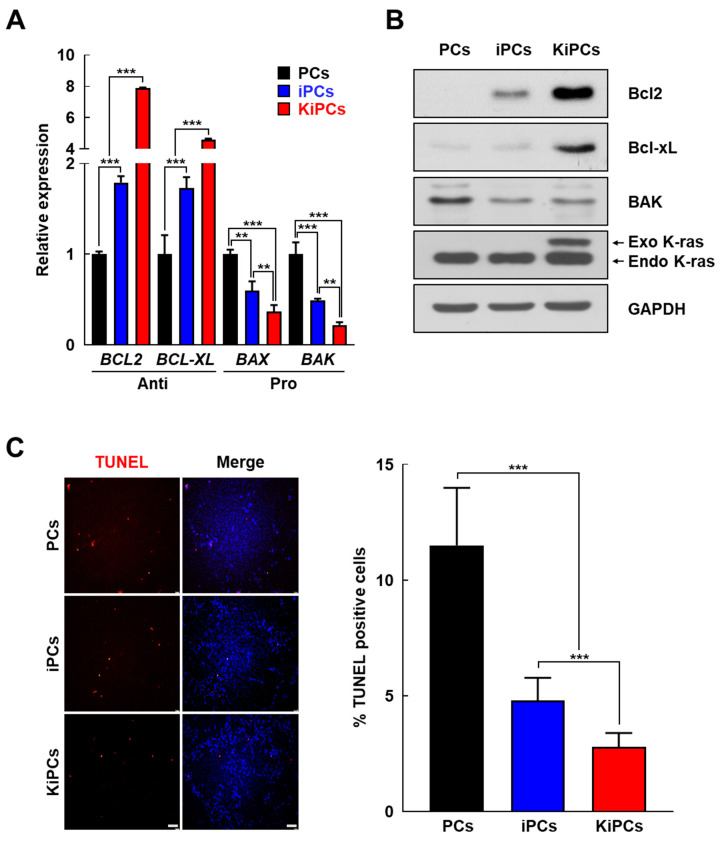
KiPCs exhibited reduced levels of apoptotic genes and Terminal deoxynucleotidyl transferase dUTP nick end labeling (TUNEL)-positive cells. (**A**) qPCR analysis of anti- and pro-apoptotic genes. (**B**) Anti- and pro- apoptotic proteins were measured by Western blot analysis. (**C**) Representative image of TUNEL assay for apoptosis (left) and the percentages of TUNEL-positive cells in the indicated groups (right). Magnification 100×. All experiments were performed using at least three replicates, and the results are representative of three independent experiments. Data are presented as means ± standard deviation. ** *p* < 0.01, *** *p* < 0.001, one-way analysis of variance with Holm–Sidak multiple comparisons.

**Figure 5 ijms-21-08820-f005:**
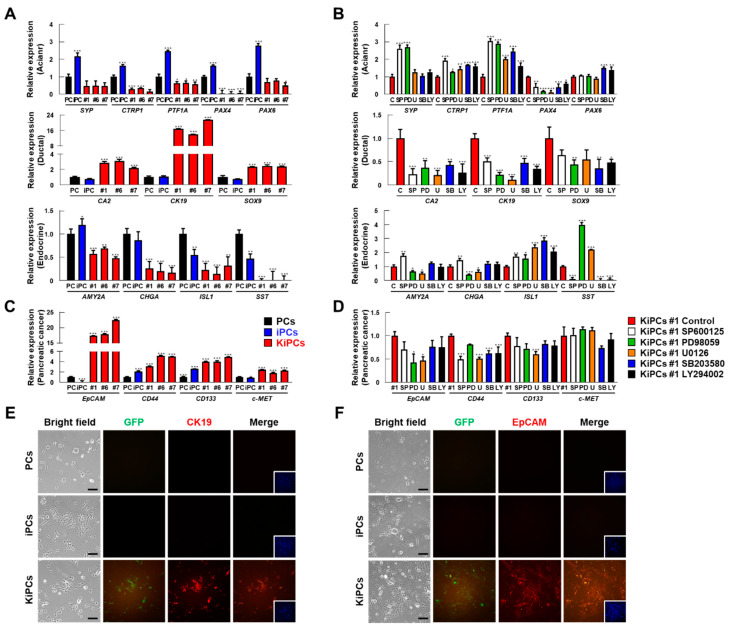
Gene expression analysis of pancreatic-specific markers and pancreatic cancer stem cell markers in KiPCs. (**A**) qPCR analysis of specific molecular markers of pancreatic acinar, ductal, and endocrine lineages in the indicated groups. (**B**) qPCR analysis of specific molecular markers of pancreatic acinar, ductal, and endocrine lineages after treatment with MAPK inhibitors in KiPCs #1. SP600125: 20 μM, PD98059: 20 μM, U0126: 10 μM, SB203580: 10 μM, LY294002: 10 μM. (**C**) qPCR analysis of specific molecular markers of pancreatic cancer markers in the indicated groups. (**D**) qPCR analysis of specific molecular markers of pancreatic cancer after treatment with MAPK inhibitors in KiPCs #1. (**E**) Double immunofluorescence was performed by staining with anti- cytoskeletal 19 (CK19; red) and anti-GFP (green) antibodies. Scale bar = 100 µm. (**F**) Double immunofluorescence was performed by staining with anti- epithelial cell adhesion molecule (EpCAM; red) and anti-GFP (green) antibodies. Scale bar = 100 µm. Magnification 100×. All experiments were performed using at least three replicates, and the results are representative of three independent experiments. Data are presented as means ± standard deviation. * *p* < 0.05, ** *p* < 0.01, *** *p* < 0.001, one-way analysis of variance with Holm–Sidak multiple comparisons.

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
