# Peer review of "Establishment and Characterization of Immortalized Miniature Pig Pancreatic Cell Lines Expressing Oncogenic K-RasG12D"

_ijms, 2020, doi:10.3390/ijms21228820_

Round 1

Reviewer 1 Report

Overall the paper is well-written and complete but there is a portion in one set of results that are not well explained. In Figure 5A, observations were made for the increased expression of acinar markers in iPCs as well as the reduction in expression of acinar markers and increased expression of ductal markers in KiPCs which was associated to ADM. However, no observation was made and explanation given  for the reduction in expression of endocrine markers in KiPCs as compared to both iPCs and PCs.

Author Response

Point 1: Overall the paper is well-written and complete but there is a portion in one set of results that are not well explained. In Figure 5A, observations were made for the increased expression of acinar markers in iPCs as well as the reduction in expression of acinar markers and increased expression of ductal markers in KiPCs which was associated to ADM. However, no observation was made and explanation given for the reduction in expression of endocrine markers in KiPCs as compared to both iPCs and PCs.

Response 1: First of all, thank you very much for your positive thoughts and good comments on our manuscript.

As reported in previous papers, endocrine cells occupy a small portion (only 1-2%) of the pancreas, and if cancer is caused by endocrine cells, they are called pancreatic endocrine neoplastic tumors (pancreatic NETs or PNETs) [1]. From our results, it is thought that the cell proliferation occurs in related target cells because KiPC cells express K-rasG12D by the acinar-specific PTF1a promoter. However, the population of endocrine cells is not only unrelated to ADM, but is expected to gradually decrease during the sequential cell line construction process due to a small portion of the pancreas. As you can see in Figure 5A, some genes of endocrine markers (ISL1 and SST) also decreased in iPCs than in PC. Nonetheless, we didn't explain it in the results section because we didn't show any results that could support them. Regarding the reviewer's good point, we will also reflect the analysis of changes in endocrine cells when evaluating pig model for pancreatic cancer in the next study.

  1. Halfdanarson, T. R.; Rubin, J.; Farnell, M. B.; Grant, C. S.; Petersen, G. M., Pancreatic endocrine neoplasms: epidemiology and prognosis of pancreatic endocrine tumors. Endocr Relat Cancer 2008, 15, (2), 409-27.

Reviewer 2 Report

Dear Authors,

Concerning the manuscript ijms-938714, entitled " Establishment and characterization of immortalized miniature pig pancreatic cell lines expressing oncogenic K-rasG12D”, is an innovative contribution for the research in pancreatic oncology.

I just recommend to (i) Describe the protocol of anesthesia and any other strategy to reduce animal suffering (Line 344) and (ii) Provide more information about the used animals, namely, the gender (Line 345)

Author Response

Point1 Concerning the manuscript ijms-938714, entitled "Establishment and characterization of immortalized miniature pig pancreatic cell lines expressing oncogenic K-rasG12D”, is an innovative contribution for the research in pancreatic oncology.

I just recommend to (i) Describe the protocol of anesthesia and any other strategy to reduce animal suffering (Line 344)

Response 1: Thank you for reviewing our manuscript very positively. We hope that our research will provide new values and innovative research directions that can be helpful in the treatment of human pancreatic cancer.

As the reviewer noted, we supplemented the anesthesia process that can reduce animal suffering in the materials and methods section of the revised manuscript.

(Omission) … All procedures and use of miniature pigs were approved by the Korea Research Institute of Bioscience and Biotechnology (KRIBB) Institutional Animal Care and Use Committee (Approval No. KRIBB-AEC-16075). All surgical procedures were performed under anaesthesia, and all efforts were made to minimize animal suffering. For euthanasia, pigs were injected with ketamine (5 mg/kg intravenously, Yuhan) followed by KCl injection (75-150 mg/kg, Dai Han Pharm. Co) and all efforts were made to minimize animal suffering by institutional veterinary experts, in accordance with NIH guidelines. A 2-week-old male KSP miniature pig … (Omission)

Point2 (ii) Provide more information about the used animals, namely, the gender (Line 345)

Response 2: We think reviewers' good comments will improve the quality of our manuscript. As mentioned in the previous our papers (PLoS One. 2019 Jul 22;14(7):e0219978, PLoS One. 2018 Oct 11;13(10):e0205495.), KSP (full name: KRIBB small pig) miniature pig, strain developed by FARRC and the gender of the pig we used is male. This part will also explain the materials and methods section.

(Omission) … For euthanasia, pigs were injected with ketamine (5 mg/kg intravenously, Yuhan) followed by KCl injection (75-150 mg/kg, Dai Han Pharm. Co) and all efforts were made to minimize animal suffering by institutional veterinary experts, in accordance with NIH guidelines. A 2-week-old male KSP miniature pig [45] was provided by the Futuristic Animal Resource & Research Center in KRIBB, Korea. Primary pancreatic cells (PCs) were prepared from the miniature pig pancreas as preciously described [46] and maintained in RPMI 1640 medium (#11875119, GIBCO) supplemented with 10% fetal bovine serum (FBS, #16000-044, GIBCO), 100 U/mL penicillin and 100 µg/mL streptomycin (#15140122, GIBCO). … (Omission).